**Data Availability Statement:** All relevant data are within the manuscript and its Supporting Information files.

**Funding:** The work is supported by the HENGRUI Foundation of Jiangsu Pharmaceutical Association

# Construction and practice of a novel pharmaceutical health literacy intervention model in psychiatric hospital

**Linghe Qiu**[1,2☯], **Jun Li**[3☯], **Weiming Xie**[1,2], **Fei Wang**[1,2], **Yuan Shen**[1,2]*, **Jianhong Wu**[1,2]*

**1** The Affiliated Mental Health Center of Jiangnan University, Wuxi, China, **2** Wuxi Central Rehabilitation Hospital, Wuxi, Jiangsu, China, **3** Nanjing First Hospital, Nanjing Medical University, Nanjing, Jiangsu, China

☯ These authors contributed equally to this work.
* 958300721@qq.com (YS); jianhong_wu@qq.com (JW)

## Abstract

### Objective

Pharmaceutical health literacy intervention (PHLI) plays a crucial role in influencing patients' medical decision-making, particularly concerning medication use. However, PHLI has not been widely implemented in China. This study aims to develop a novel PHLI model within a psychiatric hospital setting and evaluate its effectiveness.

### Methods

A PHLI model encompassing four modes—covering inpatients, outpatients, Internet+ and community—was established at The Affiliated Mental Health Center of Jiangnan University. The model's operation was detailed, and its performance data from 2022 and 2023 were evaluated.

### Results

In 2022 and 2023, a total of 636 PHLI cases were reported. Of these, 386 cases (60.69%) were identified through the inpatient mode. The proportion of PHLI delivered via inpatient and Internet information subscription modes gradually increased, while interventions through other methods decreased. The age group of 18–30 accounted for 21.97% of cases, with 116 instances reported. Various types of PHLI were provided, including adverse reactions (18.87%), dosage and administration (11.64%), and therapeutic drug monitoring (9.43%). In addition, intervention strategies primarily focused on adverse reaction identification (10.22%), interpretation of pharmaceutical reports (7.23%), and routine examination reminders (6.45%).

### Conclusion

The PHLI model developed at our hospital offers an effective approach to health literacy intervention and represents an innovation advancement in pharmaceutical health literacy management. It can also serve as a reference framework for other hospitals.

(No. H202139). The funders had role in study design of the manuscript.

## 1.Introduction

In October 2020, the National Health Commission issued "Opinions on Strengthening and Improving Psychiatric Medical Services", which emphasized the importance of enhancing the psychiatric medical system, building a comprehensive service network, and accelerating pharmacy transformation [1]. As part of this effort, pharmaceutical health literacy intervention (PHLI) plays a crucial role in improving patients' ability to understand drug information and make informed choices. In psychiatric settings, it is equally important to improve the willingness of patients with potential mental illness to seek treatment [2].

The concept of PHLI was first introduced by D. K. Theo Raynor, aiming to provide better medication-related services for populations with low health literacy [3]. Effective communication of simple and understandable medical information, both in writing and orally, is crucial for treatment success, especially when explaining medication use to discharged patients [4]. Sauceda et al. defined pharmaceutical health literacy as the ability to critically acquire, understand, and use basic medication information, thereby reducing medication errors arising from misunderstandings [5]. Building on this, Pouliot et al. refined the definition through an international expert consensus using the Delphi method. This refined definition emphasizes the ability to obtain, understand, communicate, calculate, and process specific medication information, enabling individuals to make informed decisions about their medication and health, thereby ensuring safe and effective medications use [6].

PHLI measures typically include teach-back methods, the use of simple language, and the chunk-check technique. The term "chunk-check", commonly used in the IT field, refers in health literacy to breaking down complex medical information into manageable chunks and gradually checking patient comprehension [7]. It includes the following steps: breaking down complex information into chunks, evaluating the reasonableness of the chunks, and assessing the effectiveness of information delivery. With advancements in technology, PHLI increasingly relies on medical audiovisual tools and digital medication therapy management [8]. These technological innovations enable more interactive and engaging educational approaches, such as the use of multimedia content to explain complex drug regimens, which is beneficial for patients with varying levels of health literacy and cognitive function.

A review of 72 articles involving population-wide surveys in Australia highlighted that health literacy intervention can significantly enhance the understanding of health information and recommendations among individuals with low health literacy, thereby improving treatment outcomes [9]. Moreover, numerous studies have confirmed that psychiatric health literacy intervention can improve treatment outcomes and alleviate symptoms [10]. During the hospitalization of patients with mental illness, implementing health literacy interventions, including PHLI, is both convenient and effective. Additionally, PHLI broadens the scope of pharmaceutical care.

A lack of PHLI can significantly reduce medication adherence, particularly among psychiatric patient post-discharge, leading to uncertainties in treatment management and a substantial increase in the recurrence of mental illness. While PHLI have been explored globally, such as using visual aids for illiterate patients with prescription instructions [11], they are not directly transferable to China due to differences in medical systems. China's health literacy intervention began later than those in other countries, with varying patient health literacy levels and no unified practice. Additionally, disparities in pharmaceutical expertise, pharmacist availability, and Internet penetration have hindered the development of comprehensive PHLI in psychiatric care. Currently, PHLI in China relies mainly on community-based efforts like free clinics and lectures, but a more proactive, integrated approach is needed throughout the patient care process.

A multicenter study conducted in mainland China found that caregivers of children are highly concerned about medication safety and are willing to learn about related topics [12]. Similarly, a study conducted in Finland and Malta identified a paradox where patients have low health literacy but a strong desire to participate in treatment decisions [13]. An observational study further demonstrated that elderly asthma patients wish to understand medication information but often struggle to obtain appropriate, personalized guidance [14]. Cross-sectional studies involving populations such as those with hypertension [15], outpatient care institutions [16], dialysis patients [17], and individuals with coronary heart disease [18] consistently indicate low levels of pharmaceutical health literacy. These findings underscore a critical challenge for pharmacists: bridging the gap between the specialized nature of pharmaceutical services and the general public's understanding. Since the determination of treatment plans requires informed decisions from both patients and their families, good health literacy is essential for making well-informed medical choices. Therefore, improving pharmaceutical health literacy is a crucial strategy for enhancing patient decision-making and treatment outcomes.

Most published articles focus on the development of pharmaceutical literacy assessment tools [19], the measurement of pharmaceutical health literacy [20], the role of pharmaceutical health literacy in preventing chronic diseases and influencing treatment adherence [21], and pharmaceutical health literacy education. However, to our knowledge, no studies have addressed the practical implementation of pharmaceutical health literacy. Additionally, pharmaceutical health literacy reflects patients' medication capabilities and attitudes to some extent [6]. More importantly, it facilitates a shift in the patient's role in healthcare—from mere complying with expert guidelines to making informed decisions [22]. Therefore, it is essential to actively promote pharmaceutical health literacy through practical and modular approaches.

A survey on the mental health literacy of the Chinese population from 1997 to 2018 concluded that more high-quality health literacy interventions are necessary to improve health literacy and promote positive changes in medical behavior in China [23]. This need is particularly critical for vulnerable groups, such as patients with mental illnesses and their caregivers, for whom feasible PHLI measures are currently lacking. Therefore, developing a PHLI model for psychiatric hospital is essential to provide technical support for these interventions and improve medical outcomes. Building on previous practical experience, our hospital, a key psychiatric institution in the Yangtze River Delta region, established a PHLI model in 2022 with the aim of creating a comprehensive PHLI workflow and enhancing the quality of pharmaceutical care.

## 2.Methods

### 2.1 Organizational structure design

PHLI primarily targets patients with mental illness and their caregivers, while also actively engaging the general population. To support this initiative, an interdisciplinary team comprising the pharmacy department, medical department, mental illness control center, nursing department, and community committee was established, leveraging the hospital's information system. The model included the participation of five clinical pharmacists, all of whom had completed clinical pharmacist training accredited by the National Health Commission and received certification. To guide the implementation of PHLI, the "Standard Operating Procedure for Pharmaceutical Management and Pharmaceutical Services at The Affiliated Mental Health Center of Jiangnan University" was developed. Prior to launching the intervention model, targeted training in health literacy interventions—covering essential methods such as teach-back, chunk-check, and the use of simple language—was provided to ensure

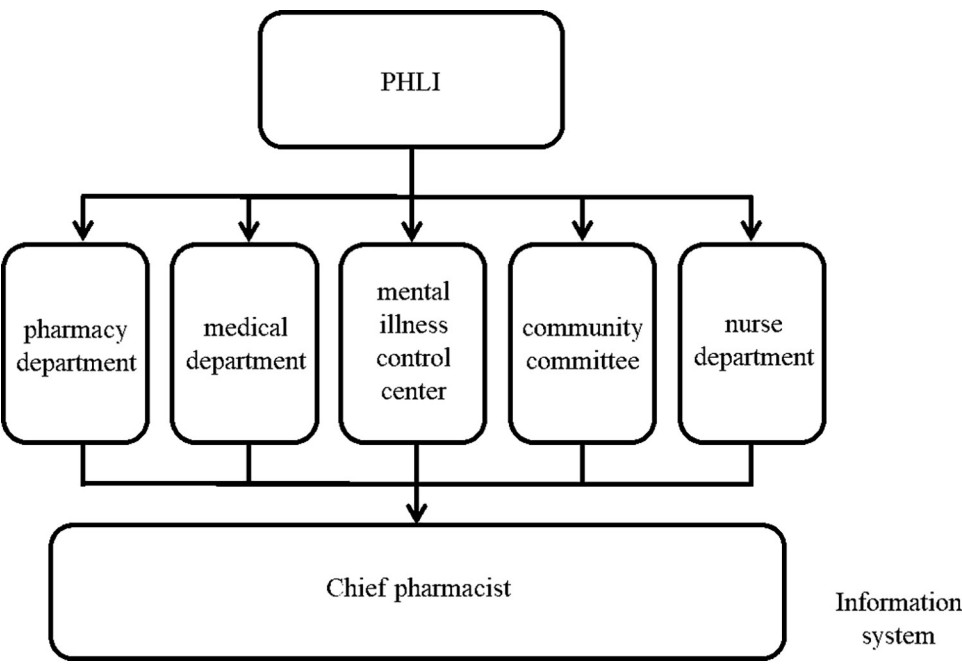

**Fig 1. Organizational structure of PHLI model.**

standardization and accuracy. All personnel reported to the chief pharmacist, and the organizational structure is illustrated in Fig 1.

## 2.2 Intervention model design

**2.2.1 Inpatient-based PHLI mode.**   The inpatient-based PHLI mode structured into four main parts. The first part involved medication reconciliation and health literacy intervention at the time of admission. Pharmacists conducted medication reconciliation, explained common adverse reactions, and provided detailed information on the function, usage, and dosage of each medication. For patients with low health literacy, PHLI was administered weekly, focusing on drug effects, proper medication methods, and how to response to adverse reactions. Additionally, real-time documentation was maintained through the internal file system, including records of pharmaceutical care, medication use, and pharmaceutical ward rounds.

The second part was the intervention during hospitalization. Throughout the treatment period, pharmacists addressed patients' questions about prescription. For patients with special conditions—such as a history of drug abuse, refusal to take medication, or a history of adverse reactions—pharmacists conducted targeted interventions and reported these cases to the chief pharmacist. Following consultation, individualized intervention strategies were developed.

The third part focused on patient discharge. After assessing the patient's discharge medication, pharmacists created an educational list detailing the diagnosis, dosage and administration instructions, potential adverse reactions, and corresponding countermeasures. This list provided targeted education for patients with multiple conditions, assisting them and their families in developing a post-discharge pharmaceutical plan to ensure continuity and stability of treatment.

The fourth part involved a retrospective analysis of the hospital information systems (HIS). HIS is a comprehensive platform for hospital information management in China, where all patient information and medical data are stored. Through HIS, after excluding patients

currently under intervention, those older than 65 years, with multiple diseases, or with more than five prescriptions were selected for focused re-rounds to confirm PHLI, particularly in areas such as medication adherence, prescription complexity, and adverse reactions. Other inpatients were screened monthly.

**2.2.2 Outpatient-based PHLI mode.** The outpatient-based PHLI mode was divided into two parts. The first part involved interventions through pharmacist-managed clinics (PMC). PMC are outpatient clinics led by clinical pharmacists, providing services such as PHLI, medication therapy management, drug consultation, medication education, and guidance on medication safety. During the medication dispensing process, pharmacists address patients' pharmaceutical concerns in real time. For patients with specific needs, pharmacists may introduce them to the PMC for targeted intervention. Clinical pharmacists in the PMC create pharmaceutical profiles for patients, recording details such as gender, age, diagnosis, medical history, allergies, and drug usage and dosage. After addressing the patients' concerns and taking necessary actions, a follow-up appointment is scheduled to ensure medication compliance.

The second part focused on interventions based on therapeutic drug monitoring (TDM) and genetic testing. Pharmacists provided specific interventions, including explanations of professional terminology, the significance of TDM, and the implications of genetic testing results on treatment. Patients with abnormal results, such as medication concentrations exceeding the warning levels or unique genotypes, were referred to the PMC to prevent serious non-compliance due to insufficient health literacy (Fig 2).

**2.2.3 Internet+ based PHLI mode.** This mode comprised three main parts. The first part involved interventions through Internet information subscription via WeChat public accounts. WeChat public accounts are the most widely used information subscription platforms in China, with nearly all medical institutions having established accounts for appointment and information dissemination. Patients can easily access hospital information, learn about expert teams, receive health services, make appointment bookings, and complete online consultations as well as payment and reimbursement services through the WeChat public account. In addition, hospitals use these accounts to publish popular science articles, authored by various professionals including doctors, pharmacists, nurses, laboratory technicians, and psychologists, to educate patients and the public in simple, understandable language. Articles by pharmacists focus on PHLI topics, such as the importance of TDM for patients with mental health disorders. When necessary, pharmacists may break down comprehensive pharmaceutical articles into a series of shorter articles (chunks) to cover the related knowledge thoroughly. Followers of the public account can read these articles and pose questions through comments. In response, pharmacists can create short videos, reply to comments, and generate additional public account articles based on readers' inquiries.

The second part involved interventions through an Internet hospital. With our hospital's Internet medical services, patients can receive medical advice from home. Although online consultations do not allow for face-to-face interactions, pharmacists can provide patients in need with online health literacy support. Weekly summaries of consultation queries were reviewed by pharmacists, and as a result, frequently asked questions were incorporated into the pharmacy-related popularization on the public account.

The third part involved interventions based on instant messaging software such as WeChat. While WeChat public accounts are institutional platforms, WeChat itself is a personal messaging tool. Many older adults in China, despite found online consultations challenging, use WeChat regularly. This platform enables patients to establish direct, one-on-one communication with pharmacists for PHLI. Pharmacists can review medication information and provide medication monitoring for patients with poor adherence or chronic conditions when necessary. The frequency of interactions is uncertain, as it depends on the initiation form of PHLI

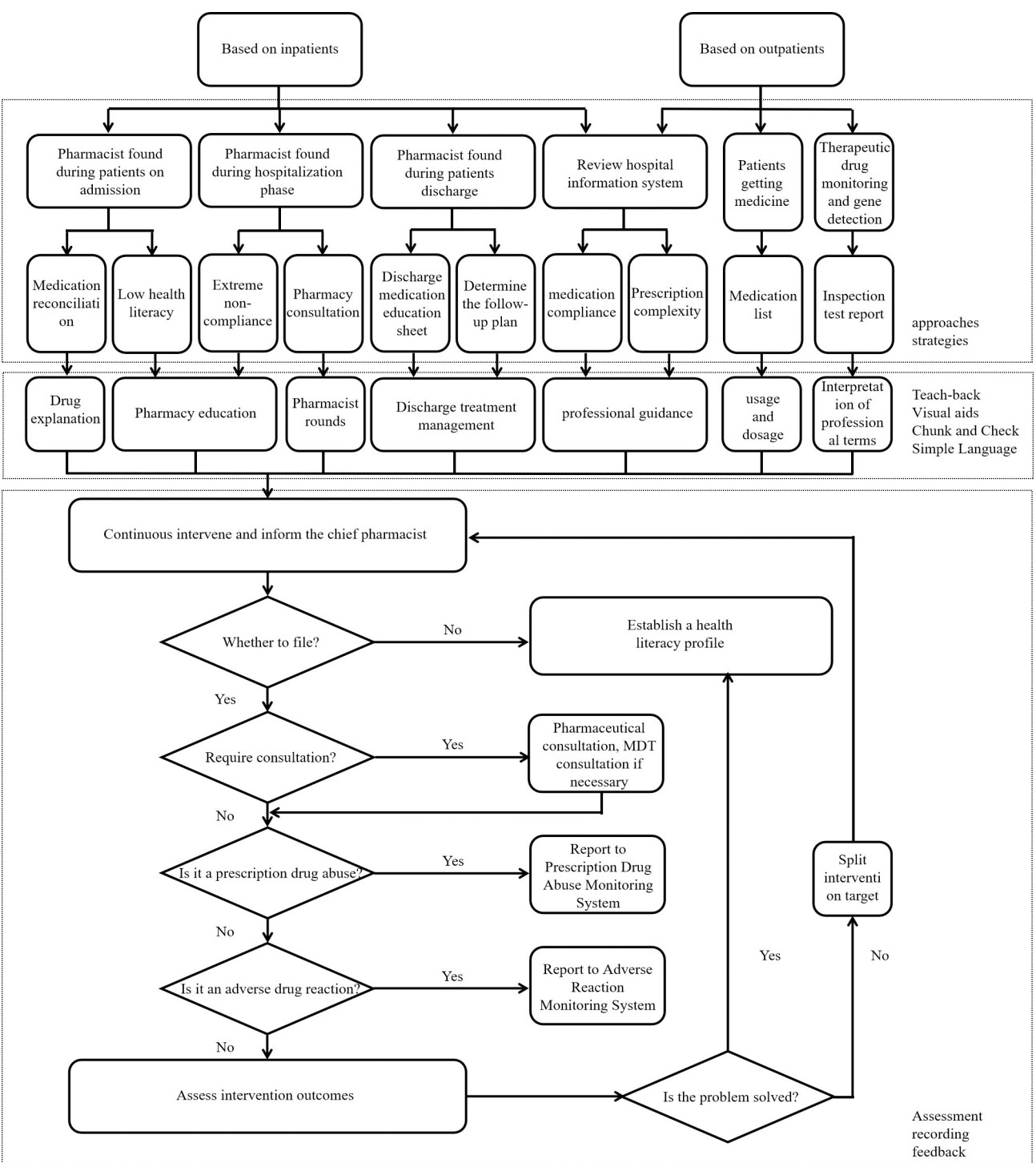

**Fig 2. Flow chart of intervention system based on inpatient and outpatient.**

and the patient's physical condition. This two-way communication simplifies interactions and allows pharmacists to dynamically monitor medication behaviors and offer real-time PHLI. More importantly, interventions based on WeChat and similar instant messaging software offer a novel perspective for addressing the dilemma of limited medical resources and improving healthcare access for older adults.

**2.2.4 Community-based PHLI mode.** The community-based PHLI mode primarily included community lectures and free clinic. Community lectures were organized through the collaborative efforts of medical associations. In China, a medical association is a consortium that brings together a large hospital and several community hospitals, characterized by shared medical resources and collaborative healthcare initiatives. As a supplement to the Internet + PHLI mode, pharmacists delivered lectures to community residents, offering one-on-one counseling sessions after the lecture to address individual questions regarding drug use, pharmaceutical knowledge, and related topics. Additionally, pharmacists assisted in managing chronic illness medications and household medicine cabinets.

Pharmacists also participated in large-scale free clinics held annually. In China, public welfare events known as free clinics are frequently organized by healthcare institutions, government bodies, and party organizations during holidays or important commemorative days. These free clinics are not limited to specific locations; they are held in various public spaces such as plazas, hospital lobbies, community centers, and nursing homes. Renowned experts provide free medical consultations and treatments to local patients, particularly those with limited access to medical care. Pharmacists involved in these free clinics conducted PHLI for individuals with low health literacy to ensure the effectiveness of their treatment (Fig 3).

## 2.3 Data statistics and analysis

The distribution of variables was assessed using descriptive statistical methods. Data management and analysis were conducted using Microsoft Excel 2016 and SPSS 24.0, respectively. Categorical data were described in terms of frequencies and percentages. Percentages were calculated by dividing the number of cases by the total number of cases for the year or the two-year period, and results were rounded to two decimal places.

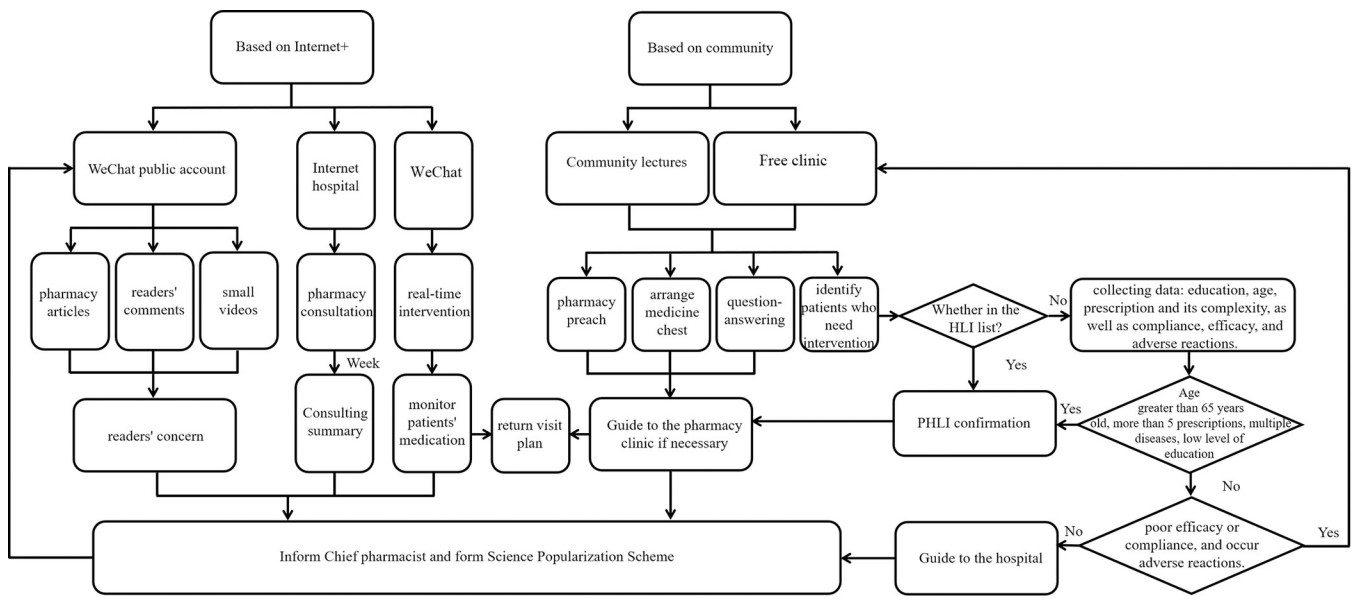

**Fig 3. Flow chart of PHLI based on Internet+ and community.**

### 2.4 Ethics approval and consent to participate

The study was conducted in accordance with the Declaration of Helsinki, and was approved by the Ethics Committee of Wuxi Mental Health Center (WXMHCIRB2023LLky007). All participants signed informed consent and the data were analyzed anonymously.

### 2.5 Data acquisition and participant information

The data was accessed on January 10, 2024, and the information of individual participants cannot be identified during or after data collection.

## 3.Results

### 3.1 Number of reported PHLI cases

Table 1 presents the number of interventions and the total patient population across different methods. Among these, 386 PHLI cases were based on inpatient care, accounting for 60.69%. PHLI through PMC identified 65 cases, representing 10.22%, while TDM and genetic testing contributed 42 cases, accounting for 6.60%. Internet information subscription uncovered 91 cases, making up 14.31%. The number of PHLI cases via Internet hospitals was 28 (4.40%), and 7 cases were identified through instant messaging software (1.10%). Additionally, 17 cases were reported through community-based PHLI, accounting for 2.67%. (See S1 Table for original data).

### 3.2 Basic information

In 2022, there were 114 PHLI cases involving males (42.07%) and 157 cases involving females (57.93%). In 2023, the number were 100 cases (38.91%) for males and 157 cases (61.09%) for females. Among both years, 200 PHLI cases involved patients aged 18 to 40, accounting for 37.88% (Table 2).

### 3.3 PHLI type distribution

The types of interventions primarily focused on adverse reactions (18.87%), dosage and administration (11.64%), and TDM (9.43%). In 2023, PHLI before pharmaceutical care accounted for 7.17% of all PHLI, with 23 cases. Patients were particularly concerned with drug usage and efficacy (Table 3).

**Table 1. The number of PHLI cases across different methods.**

| Types | 2022 | | 2023 | | Total | |
|---|---|---|---|---|---|---|
| | Cases | Ratio/% | Cases | Ratio/% | Cases | Ratio/% |
| PHLI mode based on inpatient | 165 (327) | 52.38 | 221 (378) | 68.85 | 386 (705) | 60.69 |
| PHLI mode based on PMC | 57 (63) | 18.10 | 8 (10) | 2.49 | 65 (73) | 10.22 |
| PHLI mode based on TDM and gene detection | 28 (30) | 8.89 | 14 (25) | 4.36 | 42 (55) | 6.60 |
| PHLI mode based on Internet information subscription | 35 (38) | 11.11 | 56 (59) | 17.45 | 91 (97) | 14.31 |
| PHLI mode based on Internet hospitals | 18 (22) | 5.71 | 10 (13) | 3.12 | 28 (35) | 4.40 |
| PHLI mode based on instant messaging software | 3 (5) | 0.95 | 4 (9) | 1.25 | 7 (14) | 1.10 |
| PHLI mode based on Community | 9 (19) | 2.86 | 8 (33) | 2.49 | 17 (52) | 2.67 |
| Total | 315 (504) | 100.00 | 321 (527) | 100.00 | 636 (1031) | 100.00 |

() the total number of the pool of patients.

**Table 2. Basic information of the patients.**

| | Gender | Age | | | | | | | | Total | Ratio/% |
|---|---|---|---|---|---|---|---|---|---|---|---|
| | | ≤17 | 18–30 | 31–40 | 41–50 | 51–60 | 61–70 | 71–80 | ≥81 | | |
| 2022 | Male | 11 | 27 | 20 | 20 | 16 | 15 | 5 | 0 | 114 | 42.07 |
| | Female | 20 | 47 | 25 | 10 | 26 | 22 | 3 | 4 | 157 | 57.93 |
| 2023 | Male | 9 | 16 | 19 | 12 | 16 | 10 | 13 | 5 | 100 | 38.91 |
| | Female | 37 | 26 | 20 | 19 | 13 | 23 | 10 | 9 | 157 | 61.09 |
| Total | | 77 | 116 | 84 | 61 | 71 | 70 | 31 | 18 | 528 | |
| Ratio/% | | 14.58 | 21.97 | 15.91 | 11.55 | 13.45 | 13.26 | 5.87 | 3.41 | 100.00 | |

### 3.4 PHLI strategies

The intervention strategies primarily focused on adverse reaction identification (10.22%), interpretation of pharmaceutical reports (7.23%), and routine examination reminders (6.45%). Ensuring medication compliance and managing adverse reactions were key areas of focus within these strategies (Table 4).

### 3.5 Drug distribution

Table 5 displays the top 10 most frequently distributed medications within the PHLI model.

## 4.Discussion

With the rise of individualized treatment, medical strategies have become increasingly complex, making health literacy a critical factor in patients' understanding of their care. This has garnered significant clinical attention [24]. PHLI is designed to assist patients, particularly those with low health literacy, in addressing common clinical challenges such as medication adherence, chronic disease self-management, and post-discharge medication [25]. For psychiatric patients and their caregivers, developing accurate PHLI is critical, covering aspects like medication timing, administration methods, managing non-compliance, and responding to adverse reactions [26]. However, there is currently no comprehensive solution for practical implementation. Most existing models focus on intervention measure, outcomes, and methodological development, but lack a structured operational process based on hospital practice [27,28]. Fortunately, China is increasingly prioritizing health literacy, highlighting the need to establish and enhance a PHLI network.

Our hospital was the first to implement psychiatric PHLI, with limited prior experience to draw upon, and the PHLI model remains in its early stages. Nevertheless, the model we developed facilitates easier and more convenient patient interventions. It addresses potential gaps—such as overlooking patients with low health literacy due to high workload or staff shortages—through community lectures, Internet+, instant messaging software, and retrospective information systems analysis. This comprehensive approach aims to reach nearly all patients within our hospital. Psychiatric patients often exhibit low health literacy and significant non-compliance behaviors, including irregular medication use, even hiding medications, refusing to take medications, and drug abuse [29,30]. The multi-method collaborative intervention strategy employed in our PHLI model is designed to comprehensively enhance patients' pharmaceutical health literacy. Internet+ and community-based PHLI help compensate for follow-up losses due to limited pharmacist resources and ensure a broader reach. Post-intervention, pharmacists establish patient records and provide comprehensive pharmaceutical care. Additionally, our PHLI model integrates with the hospital's adverse reaction and drug abuse monitoring systems, broadening the intervention scope through rescreening patients within these

**Table 3. PHLI type distribution of the reported cases.**

| Types | 2022 | | 2023 | | Total | |
|---|---|---|---|---|---|---|
| | Cases | Ratio/% | Cases | Ratio/% | Cases | Ratio/% |
| Adverse reactions | 66 | 20.95 | 54 | 16.82 | 120 | 18.87 |
| Dosage and administration | 33 | 10.48 | 41 | 12.77 | 74 | 11.64 |
| TDM | 39 | 12.38 | 21 | 6.54 | 60 | 9.43 |
| Special dosage form administration | 26 | 8.25 | 27 | 8.41 | 53 | 8.33 |
| Pediatric drug use | 22 | 6.98 | 19 | 5.92 | 41 | 6.45 |
| Medical science popularization | 17 | 5.40 | 30 | 9.35 | 47 | 7.39 |
| Medication compliance | 12 | 3.81 | 18 | 5.61 | 30 | 4.72 |
| Drug selection | 12 | 3.81 | 6 | 1.87 | 18 | 2.83 |
| Medication timing | 9 | 2.86 | 17 | 5.30 | 26 | 4.09 |
| Drug interactions | 8 | 2.54 | 12 | 3.74 | 20 | 3.14 |
| Treatment course | 8 | 2.54 | 1 | 0.31 | 9 | 1.42 |
| Pharmacogenomics | 7 | 2.22 | 4 | 1.25 | 11 | 1.73 |
| Drug substitution | 5 | 1.59 | 2 | 0.62 | 7 | 1.10 |
| Drug abuse | 5 | 1.59 | 2 | 0.62 | 7 | 1.10 |
| Pharmaceutical care | 4 | 1.27 | 23 | 7.17 | 27 | 4.25 |
| Discontinuation strategy | 4 | 1.27 | 8 | 2.49 | 12 | 1.89 |
| Drug onset time | 4 | 1.27 | 5 | 1.56 | 9 | 1.42 |
| Self-medication or discontinuation | 4 | 1.27 | | 0.00 | 4 | 0.63 |
| Treatment plan interpretation | 3 | 0.95 | 15 | 4.67 | 18 | 2.83 |
| Drug overdose | 3 | 0.95 | 7 | 2.18 | 10 | 1.57 |
| Patient concerns | 3 | 0.95 | 1 | 0.31 | 4 | 0.63 |
| Drug combination | 3 | 0.95 | 1 | 0.31 | 4 | 0.63 |
| Drug regimen adjustment | 3 | 0.95 | 1 | 0.31 | 4 | 0.63 |
| Pregnancy medication | 3 | 0.95 | | 0.00 | 3 | 0.47 |
| Test result interpretation | 2 | 0.63 | | 0.00 | 2 | 0.31 |
| Prescription issuance rules | 2 | 0.63 | 3 | 0.93 | 5 | 0.79 |
| Drug addiction | 2 | 0.63 | 2 | 0.62 | 4 | 0.63 |
| Medication and diet | 2 | 0.63 | 1 | 0.31 | 3 | 0.47 |
| Living habits | 2 | 0.63 | | 0.00 | 2 | 0.31 |
| Pharmaceutical compounding | 1 | 0.32 | | 0.00 | 1 | 0.16 |
| Exercise therapy | 1 | 0.32 | | 0.00 | 1 | 0.16 |
| Total | 315 | | 321 | | 636 | 100 |

system [31]. While full-time intervention pharmacists reduce communication gaps and waiting times, this approach demands significant manpower, resources and funding. Continuous improvement of the PHLI model is essential, along with the development of new interventions, such as joint physician-pharmacist interventions, chronic disease management models, and targeted interventions for the digital divide among older adults [32].

The increasing proportion of PHLI conducted via Internet information subscription, particularly through platforms like WeChat public account articles, indicates the growing significance of these novel methods. These articles effectively simplify complex information by breaking it down into digestible chunks using simple language and multimedia elements, making them highly acceptable. Digital interventions often outperform traditional didactic approaches in terms of effectiveness [33]. Additionally, the self-iterative nature of these platforms allows for ongoing refinement of article publishing strategy based on reader feedback. The predominance of young patients in PHLI may be attributed to their greater receptiveness

**Table 4. PHLI strategies distribution of the cases.**

| Types | 2022 | | 2023 | | Total | |
|---|---|---|---|---|---|---|
| | Cases | Ratio/% | Cases | Ratio/% | Cases | Ratio/% |
| Adverse reaction identification | 38 | 12.06 | 27 | 8.41 | 65 | 10.22 |
| Interpretation of pharmaceutical report | 32 | 10.16 | 14 | 4.36 | 46 | 7.23 |
| Routine examination reminders | 15 | 4.76 | 26 | 8.10 | 41 | 6.45 |
| Monitoring of drug plasma levels | 22 | 6.98 | 16 | 4.98 | 38 | 5.97 |
| Popular science on different dosage forms | 19 | 6.03 | 21 | 6.54 | 40 | 6.29 |
| Avoidance of sudden drug withdrawal | 14 | 4.44 | 11 | 3.43 | 25 | 3.93 |
| Dose reduction | 13 | 4.13 | 9 | 2.80 | 22 | 3.46 |
| Disease symptoms identification | 12 | 3.81 | 21 | 6.54 | 33 | 5.19 |
| Adverse reaction reminders | 12 | 3.81 | 22 | 6.85 | 34 | 5.35 |
| Seeking medical treatment | 11 | 3.49 | 6 | 1.87 | 17 | 2.67 |
| Weight monitoring | 11 | 3.49 | 15 | 4.67 | 26 | 4.09 |
| Adjustment of medication timing | 11 | 3.49 | 17 | 5.30 | 28 | 4.40 |
| Continuing efficacy assessment | 10 | 3.17 | 13 | 4.05 | 23 | 3.62 |
| Treatment of disease symptoms | 9 | 2.86 | 19 | 5.92 | 28 | 4.40 |
| Treatment of adverse reactions | 9 | 2.86 | 5 | 1.56 | 14 | 2.20 |
| avoidance of drug abuse | 8 | 2.54 | 6 | 1.87 | 14 | 2.20 |
| Avoidance of medication omissions | 8 | 2.54 | 19 | 5.92 | 27 | 4.25 |
| Drug titration | 7 | 2.22 | 16 | 4.98 | 23 | 3.62 |
| Avoidance of crushing medication | 7 | 2.22 | 4 | 1.25 | 11 | 1.73 |
| Popular science on drug distinctions | 6 | 1.90 | 3 | 0.93 | 9 | 1.42 |
| Alleviation of medication anxiety | 6 | 1.90 | 3 | 0.93 | 9 | 1.42 |
| Avoidance of arbitrary dose reduction | 6 | 1.90 | 2 | 0.62 | 8 | 1.26 |
| Avoidance of arbitrary dose escalation | 6 | 1.90 | 4 | 1.25 | 10 | 1.57 |
| Avoidance of long-term use | 5 | 1.59 | 2 | 0.62 | 7 | 1.10 |
| Consultation at a PMC | 3 | 0.95 | 3 | 0.93 | 6 | 0.94 |
| Avoidance of simultaneous use | 3 | 0.95 | 2 | 0.62 | 5 | 0.79 |
| Resumption of medication | 2 | 0.63 | | 0.00 | 2 | 0.31 |
| Alcohol control | 2 | 0.63 | | 0.00 | 2 | 0.31 |
| Simultaneous administration | 2 | 0.63 | 1 | 0.31 | 3 | 0.47 |
| Regular exercise | 2 | 0.63 | 1 | 0.31 | 3 | 0.47 |
| Avoidance of arbitrary medication changes | 2 | 0.63 | | 0.00 | 2 | 0.31 |
| Modification of drug type | 1 | 0.32 | 3 | 0.93 | 4 | 0.63 |
| Avoidance of mixing drugs | 1 | 0.32 | 2 | 0.62 | 3 | 0.47 |
| Popular science on treatment plan distinctions | | 0.00 | 3 | 0.93 | 3 | 0.47 |
| Implementation of dietary therapy | | 0.00 | 1 | 0.31 | 1 | 0.16 |
| Drug poisoning identification | | 0.00 | 4 | 1.25 | 4 | 0.63 |
| Total | 315 | | 321 | | 636 | 100.00 |

to diverse forms of intervention and their need for higher health literacy to balance efficacy with adverse reactions. Adverse reactions remain a primary focus in PHLI, as they allow for direct identification and timely intervention of patient issues. Ensuring medication compliance and providing simple pharmaceutical guidance are central to PHLI efforts.

The PHLI model developed in this study offers innovative approaches to addressing health literacy gap and provides more targeted interventions for patients. This model also contributes to reducing patient health management risks and is highly accessible. Its high replicability and reliance on routine technologies, which do not incur additional costs, suggest that it could be

**Table 5. Top 10 most frequently distributed medications in reported cases.**

| Classification | Cases | Ratio/% | Name | Cases | Ratio/% |
|---|---|---|---|---|---|
| antipsychotic drug | 312 | 45.28 | Olanzapine | 152 | 22.06 |
| | | | Quetiapine | 120 | 17.42 |
| | | | Aripiprazole | 40 | 5.81 |
| sedative-hypnotics | 172 | 24.96 | Lorazepam | 64 | 9.29 |
| | | | Oxazepam | 57 | 8.27 |
| | | | Zopiclone | 51 | 7.40 |
| antiepileptic drugs | 92 | 13.35 | Sodium valproate | 92 | 13.35 |
| antidepressant | 70 | 10.16 | Escitalopram | 36 | 5.22 |
| | | | sertraline | 34 | 4.93 |
| Anti-mania drugs | 43 | 6.24 | Lithium carbonate | 43 | 6.24 |
| Total | 689 | 100.00 | | 689* | 100.00 |

* The total number of medications exceeds 636 because some people may take more than one medication at the same time.

effectively implemented and refined in various countries and health systems [31]. However, there are notable limitations, including the lack of health literacy classification, insufficient publicity, and variability in pharmacists' professional skills. Follow-up research will further improve the PHLI model and incorporate more special intervention projects, expend the coverage of intervention, raise awareness of pharmaceutical care, optimize continuing education mechanisms, and establish a robust foundation for subsequent policy development.

## 5.Conclusion

The PHLI model implemented at our hospital represents an innovation approach to health literacy management, enhancing both the operation and scope of PHLI. This model effectively addresses the increasing health literacy needs of the public, helps to narrow the health literacy gap, and enables timely intervene in the early stages of clinical practice. It also contributes to the provision of diverse clinical pharmaceutical services.

## Supporting information

**S1 Table.**
(XLS)

## Author Contributions

**Conceptualization:** Yuan Shen.

**Data curation:** Weiming Xie, Jianhong Wu.

**Formal analysis:** Linghe Qiu, Jun Li, Weiming Xie.

**Funding acquisition:** Jianhong Wu.

**Investigation:** Fei Wang.

**Methodology:** Jun Li, Fei Wang, Jianhong Wu.

**Project administration:** Linghe Qiu.

**Resources:** Weiming Xie, Jianhong Wu.

**Software:** Fei Wang.

**Supervision:** Jun Li, Yuan Shen.

**Validation:** Linghe Qiu.

**Writing – original draft:** Linghe Qiu.

**Writing – review & editing:** Jun Li, Jianhong Wu.

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
