## [Decision Letter · Decision Letter 0]

9 Aug 2024

PONE-D-24-14994Construction and practice of a novel pharmaceutical health literacy intervention model in psychiatric hospitalPLOS ONE

Dear Dr. Wu,

Thank you for submitting your manuscript to PLOS ONE. After careful consideration, we feel that it has merit but does not fully meet PLOS ONE’s publication criteria as it currently stands. Therefore, we invite you to submit a revised version of the manuscript that addresses the points raised during the review process.

We look forward to receiving your revised manuscript.

Kind regards,

Yong-Hong Kuo

Academic Editor

PLOS ONE

“The work is supported by the HENGRUI Foundation of Jiangsu Pharmaceutical Association (No. H202139).”

Additional Editor Comments:

Based on the referees' recommendations and comments, I recommend Major Revision.

In addition to the referees' suggestions, please:

1. Have a more comprehensive literature review to summarize the previous research and support the current research

2. Seek professional editing services to enhance the readability of the work.

Reviewers' comments:

Reviewer's Responses to Questions

**Comments to the Author**

1. Is the manuscript technically sound, and do the data support the conclusions?

Reviewer #1: Partly

Reviewer #2: Yes

2. Has the statistical analysis been performed appropriately and rigorously? 

Reviewer #1: N/A

Reviewer #2: Yes

3. Have the authors made all data underlying the findings in their manuscript fully available?

Reviewer #1: No

Reviewer #2: Yes

4. Is the manuscript presented in an intelligible fashion and written in standard English?

Reviewer #1: Yes

Reviewer #2: Yes

5. Review Comments to the Author

Reviewer #1: The term “chunk-check” can be more or less used in the software, programming and IT fields, but we are not aware of a specific meaning in the health field. The authors should therefore check its use and explain it or replace it with a more widely understood term.

“Pharmacists in pharmacy clinic create pharmaceutical files….”. We do not know whether this is a mistranslation to English, but it is not clear to us what is a “pharmacy clinic”. Similarly, we do not know what is a “public welfare free clinic”.

We find very interesting the WeChat platform, it would be very useful if the authors could expand a bit its description for the uninformed readers.

“The third was intervention based on instant messaging software. Many elderly people in China found it challenging to conduct online consultations with pharmacies. As a result, pharmacists can communicate via instant messaging software with patients, allowing them to monitor the patients’ medications and provide real-time dynamic PHLI as needed.” This is actually very vague: what instant messaging software was used? How where the patients having access to the pharmacists for messaging? How frequent were the interactions?

In the Results section, the authors only provide numbers of patients on whom one intervention was carried out. However, it would be useful to know what was the total number of the pool of patients out of whom those interventions were applied (i.e. how frequent was a PHLI intervention among inpatients? how frequent was among outpatients? Etc).

“Among them, 200 cases ranged from 18 to 40 years old, accounting for 37.88 % (Table 2).” “Among them” refers to patients from 2022, from 2023 or the total number of the two years?

“Among them, pharmaceutical care accounted for 7.17 % of PHLI in 2023, with 23 cases.” How was pharmaceutical care defined in this context?

A number of the Types listed in Table 3 need clarification for the interested readers: what does “Usage of special dosage forms” refer to? “Medical science popularization“? “Drug gene”? (what was the pharmacist activity relating to the “drug genes”?) “Pharmaceutical care”? “Index interpretation“? “Prescription issued“?

The same holds true for Types listed in Table 4: “Interpretation of Pharmaceutical Report“? “Regular inspections”? “Avoid grinding to take”? “Drug difference”? “Avoid random reduction“? “Avoid random additions”? “Active movement“? “The difference of treatment plan”?

It would have been useful to also know how many pharmacists were involved in those interventions. Was there any additional training before starting the interventions? Was there any guidance/documentation available for pharmacists to ensure consistent and uniform provision of the interventions? What would be the average time for an intervention? (possibly by types of interventions).

Moderate changes of English necessary (agreements between nouns and verbs etc).

Reviewer #2: I think authors need to include more information in the introduction and explain more about statistical analysis they used. English language was good.

6. PLOS authors have the option to publish the peer review history of their article (what does this mean?). If published, this will include your full peer review and any attached files.

Reviewer #1: No

Reviewer #2: No

---

## [Author Response · Author response to Decision Letter 0]

11 Sep 2024

Dear editor/reviewers,

We apologize for the delayed response. We greatly appreciate your constructive feedback and professional advice. These suggestions have helped us enhance the academic rigor of our article. We have thoroughly revised the manuscript in response to the comments from the editors and reviewers.

In the Introduction section, we included a literature review, refined the explanation of the "chunk-check" method, and thoroughly reorganized the structure. In the Methods section, we improved the content regarding pharmacists' qualifications and training, introduced the use of WeChat public account and instant messaging software (WeChat), and provided a more detailed overview of the statistical analysis methods. In the Results section, we increased the patient pool for PHLI and revised unclear statements.

Finally, we apologize for the poor language quality and have conducted a thorough language revision. We have uploaded the marked manuscript, with the page numbers in parentheses corresponding to the simplified revision mode.

Additional Editor Comments:

Based on the referees' recommendations and comments, I recommend Major Revision.

We sincerely appreciate the opportunity to revise our manuscript. Following your suggestions, we have improved the literature review and made enhancements to the language.

In addition to the referees' suggestions, please:

1. Have a more comprehensive literature review to summarize the previous research and support the current research

We greatly appreciate this constructive feedback, which has made our article more comprehensive. In the Introduction section, we have added a literature review that elaborates on the definition, development, and clinical utility of PHLI. Additionally, we have summarized the current research directions of PHLI, the challenges encountered in clinical practice, and the existing research gaps that our study aims to fill, providing a basis for our research. (Please refer to pages 4-8, lines 47-131)

2. Seek professional editing services to enhance the readability of the work.

We appreciate your constructive suggestions and have sought the assistance of a professional editor to revise the language of the manuscript.

Reviewer #1: The term “chunk-check” can be more or less used in the software, programming and IT fields, but we are not aware of a specific meaning in the health field. The authors should therefore check its use and explain it or replace it with a more widely understood term.

Thank you for your kind reminder. Indeed, as you mentioned, the term "chunk-check" is more commonly used in the IT field, where it refers to breaking down large tasks into specific smaller chunks and checking each chunk sequentially. Now, this concept is also being applied to other fields. In the realm of health literacy, the application of "chunk-check" involves breaking down complex health information into smaller, more manageable parts and gradually checking for understanding to ensure the information is conveyed clearly and is easy to understand. [1] It helps to enhance the health decision-making capabilities of the audience.

Health literacy focuses on an individual's ability to obtain, understand, and use health information. Therefore, when dealing with health-related content, chunk-check method helps improve the effectiveness of information delivery. [2]

In this context, chunk-check typically involves the following aspects:

1. Chunking Information: Breaking down complex health information into smaller, more understandable segments. For example, dividing a complicated pharmaceutical explanation into shorter sentences or sections, allowing patients to absorb the information gradually.

2. Evaluating the Reasonableness of Chunks: Assessing whether these chunks are appropriate for the target audience, particularly the comprehension levels of psychiatric patients and their families. This includes checking sentence length, paragraph structure, word difficulty, and the logical flow and coherence of the information conveyed.

3. Assessing the Effectiveness of Information Delivery: Testing the patient’s understanding of the chunked health information through questioning and the teach-back method, and adjusting any content that is unclear or potentially misleading.

For instance, when writing instructions on titration therapy for depression patients, the chunk-check process involves dividing the instructions into different sections (such as definition, method, and efficacy observation) and using short sentences, clear steps, and concise language within each section. Then, we check if these sections are logical, appropriate for the reading levels of specific depression patients (e.g., those under 13 years old or over 65 years old), and if the educational objectives are met.

We sincerely appreciate your kind reminder, and we have added this explanation to the text. (Please refer to pages 5, lines 60-70)

“Pharmacists in pharmacy clinic create pharmaceutical files….”. We do not know whether this is a mistranslation to English, but it is not clear to us what is a “pharmacy clinic”. Similarly, we do not know what is a “public welfare free clinic”.

We apologize for the unclear expression. A more appropriate term than "pharmacy clinic" is "pharmacist-managed clinics" (PMC). [3] In China, PMC is not a standalone clinic but a specialized department within a hospital that provides outpatient pharmaceutical services. PMC is a healthcare method led by clinical pharmacists, aiming to offer services such as PHLI, medication therapy management, drug consultation, medication education, and medication safety guidance. The main responsibilities of PMC include: personalized medication plan development and optimization, monitoring and management of adverse drug reactions, polypharmacy management, chronic disease management, and patient medication education. This service model not only enhances patient medication safety and efficacy but also alleviates the burden on doctors and promotes collaborative healthcare. (Please refer to pages 11, lines 189-192)

"Public welfare free clinic" refers to free clinics organized by healthcare institutions, government bodies, or party organizations. In China, these free clinics are frequently held during holidays or significant commemorative days. They are not located in specific clinics but are set up in various locations such as plazas, hospital lobbies, community centers, and nursing homes. Renowned experts provide free medical consultations and treatments to local patients. Pharmacists also participate in these events, providing pharmaceutical health literacy interventions. We have already revised it in the manuscript to "free clinics." (Please refer to pages 15, lines 261-267)

Thank you for your reminder. We have improved the language expression and added explanatory notes.

We find very interesting the WeChat platform, it would be very useful if the authors could expand a bit its description for the uninformed readers.

We greatly appreciate your constructive suggestions, and we are pleased to see your interest in the WeChat platform. We have refined the discussion regarding it.

The WeChat is the largest mobile social platform in China and WeChat public account is an information subscription platform launched by it. Almost all medical institutions have established public accounts on WeChat. Due to the experience of internet-based appointment during the COVID-19 pandemic, patients in China have developed the habit of subscribing to hospital information through WeChat. Hospital WeChat public accounts mainly provide three types of content: online appointment and payment, dissemination of medical information, and publication of popular science articles. Patients can subscribe to articles about specific content related to the hospital on WeChat (The following image is a specific interface). 

Research shows that the WeChat platform can be an effective intervention to alleviate anxiety in patients or caregivers, [4] and can also enhance the effectiveness of health literacy interventions, [5] and treatment outcomes. [6] Among these, popular science articles on WeChat play a significant role in patient health literacy interventions. Many healthcare professionals, including doctors, pharmacists, nurses, laboratory technicians, and counselors, publish popular science articles on WeChat public accounts, explaining diseases, medications, or treatments in simple language to improve public health literacy. Currently, articles on WeChat not only include text but also images (comics), videos, and games, which greatly promote pharmaceutical health literacy among the public (including patients). Short, independent popular science articles can be considered an important form of "chunk-check". As shown in the results section, the proportion of our hospital’s Internet information subscription mode in PHLI is steadily increasing, indicating that innovative PHLI methods such as WeChat popular science articles are playing an increasingly important role.

In those popular science articles, pharmacists break down complex pharmaceutical knowledge into chunks, using simple language and images to simplify difficult-to-understand information. For example, when explaining the relationship between pharmacogenomics and depression treatment, pharmacists often start with a case that highlights varying responses to the same antidepressant among different patients, pointing out the potential role of pharmacogenomics. Next, pharmacists use images or videos to explain medical terms such as pharmacogenomics, genotype, and cytochrome P450 enzyme. They then describe the definition of metabolism and how different genotypes affect it. Finally, pharmacists emphasize how genetic testing determines a patient's genotype to guide medication use. These articles are published as a series, and comments below the articles are collected. Pharmacists adjust the plans of the articles based on those comments to address information gaps in a timely and accurate manner, completing the self-iterative function of the internet information subscription model.

Similar to your feelings, we also found that WeChat platform plays a unique role in PHLI. Thank you again for this constructive suggestion. (Please refer to pages 12-13, lines 212-227)

“The third was intervention based on instant messaging software. Many elderly people in China found it challenging to conduct online consultations with pharmacies. As a result, pharmacists can communicate via instant messaging software with patients, allowing them to monitor the patients’ medications and provide real-time dynamic PHLI as needed.” This is actually very vague: what instant messaging software was used? How where the patients having access to the pharmacists for messaging? How frequent were the interactions?

We apologize for any unclear statements and have refined this section to better address your concerns. Instant messaging software in our context primarily refers to WeChat. In China, WeChat functions both as an institutional platform for publishing and subscribing to information and as an instant messaging tool for personal communication. While some older adults may struggle with reading WeChat public account articles or navigating complex online medical interfaces, they primarily use WeChat for personal communication. We find that using WeChat allows us to reach this demographic, overcoming spatial limitations and delivering high-quality PHLI to improve outcomes. Through WeChat, patients can establish one-on-one contact with pharmacists. If necessary, pharmacists can access patients' medication information and provide monitoring. Therefore, if the PHLI is initiated passively by the patient, it often does not follow a fixed frequency and pharmacists providing immediate PHLI services as needed. However, if the pharmacist actively initiates the monitoring service for patients, interactions are conducted daily. (Please refer to pages 13-14, lines 237-244)

In the Results section, the authors only provide numbers of patients on whom one intervention was carried out. However, it would be useful to know what was the total number of the pool of patients out of whom those interventions were applied (i.e. how frequent was a PHLI intervention among inpatients? how frequent was among outpatients? Etc).

We appreciate your professional suggestion and have added the total number of the pool of patients in parentheses after each case in Table 1. The details are as follows: (Please refer to pages 17, Table 1)

Table 1 The number of PHLI cases in different ways

Types 2022 2023 Total 

 Cases Ratio/% Cases Ratio/% Cases Ratio/%

PHLI mode based on inpatient 165 (327) 52.38 221 (378) 68.85 386 (705) 60.69 

PHLI mode based on pharmacist-managed clinics 57 (63) 18.10 8 (10) 2.49 65 (73) 10.22 

PHLI mode based on TDM and gene detection 28 (30) 8.89 14 (25) 4.36 42 (55) 6.60 

PHLI mode based on Internet information subscription 35 (38) 11.11 56 (59) 17.45 91 (97) 14.31 

PHLI mode based on Internet hospitals 18 (22) 5.71 10 (13) 3.12 28 (35) 4.40 

PHLI mode based on instant messaging software 3 (5) 0.95 4 (9) 1.25 7 (14) 1.10 

PHLI mode based on Community 9 (19) 2.86 8 (33) 2.49 17 (52) 2.67 

Total 315 (504) 100.00 321 (527) 100.00 636 (1031) 100.00 

“Among them, 200 cases ranged from 18 to 40 years old, accounting for 37.88 % (Table 2).” “Among them” refers to patients from 2022, from 2023 or the total number of the two years?

We apologize for the unclear expression. The 200 cases refer to the total number of 2022 and 2023, specifically the sum of the numbers highlighted in red in the following table (Table 2). The details are as follows: (Please refer to pages 17, lines 308-309)

Table 2 Basic information of the patients

 Gender Age Total Ratio/%

 ≤17 18-30 31-40 41-50 51-60 61-70 71-80 ≥81 

2022 Male 11 27 20 20 16 15 5 0 114 42.07

 Female 20 47 25 10 26 22 3 4 157 57.93

2023 Male 9 16 19 12 16 10 13 5 100 38.91

 Female 37 26 20 19 13 23 10 9 157 61.09

Total 77 116 84 61 71 70 31 18 528 

Ratio/% 14.58 21.97 15.91 11.55 13.45 13.26 5.87 3.41 100.00 

“Among them, pharmaceutical care accounted for 7.17 % of PHLI in 2023, with 23 cases.” How was pharmaceutical care defined in this context?

Pharmaceutical care refers to the patient-centered pharmaceutical services provided by clinical pharmacists. In China, these services include a comprehensive range of medication-related plans, such as medication reconciliation, drug therapy assessment, therapeutic drug monitoring (TDM), medication education and guidance, and adverse drug reaction management. These services are often tailored for patients on multiple medications, older adults, children, those with chronic diseases, special physiological conditions, severe comorbidities, or a history of adverse drug reactions. 

When it is determined that a patient needs medication monitoring, pharmacists simultaneously initiate PHLI. They communicate information about pharmaceutical care through various means (such as written, verbal, and visual) to help patients make informed decisions about their medication therapy and health. It is evident that in 2023, the PHLI in the field of pharmaceutical care at our hospital has increased significantly compared to 2022. We have made the corresponding revisions in the text. (Please refer to pages 18, lines 314-315)

A number of the Types listed in Table 3 need clarification for the interested readers: what does “Usage of special dosage forms” refer to? “Medical science popularization“? “Drug gene”? (what was the pharmacist activity relating to the “drug genes”?) “Pharmaceutical care”? “Index interpretation“? “Prescription issued“?

The same holds true for Types listed in Table 4: “Interpretation of Pharmaceutical Report“? “Regular inspections”? “Avoid grinding to take”? “Drug difference”? “Avoid random reduction“? “Avoid random additions”? “Active movement“? “The difference of treatment plan”?

We apologize for the poor language. Table 3 refers to the types of PHLI, specifically the distribution of fields where pharmacists engage in PHLI. This includes interventions related to adverse reaction, 

---

## [Decision Letter · Decision Letter 1]

26 Sep 2024

Construction and practice of a novel pharmaceutical health literacy intervention model in psychiatric hospital

PONE-D-24-14994R1

Dear Dr. Wu,

We’re pleased to inform you that your manuscript has been judged scientifically suitable for publication and will be formally accepted for publication once it meets all outstanding technical requirements.

Kind regards,

Yong-Hong Kuo

Academic Editor

PLOS ONE

Additional Editor Comments (optional):

All the concerns from the referees have been addressed. I recommend Accept.

Reviewers' comments:

Reviewer's Responses to Questions

**Comments to the Author**

1. If the authors have adequately addressed your comments raised in a previous round of review and you feel that this manuscript is now acceptable for publication, you may indicate that here to bypass the “Comments to the Author” section, enter your conflict of interest statement in the “Confidential to Editor” section, and submit your "Accept" recommendation.

Reviewer #1: All comments have been addressed

Reviewer #2: All comments have been addressed

2. Is the manuscript technically sound, and do the data support the conclusions?

Reviewer #1: Yes

Reviewer #2: Yes

3. Has the statistical analysis been performed appropriately and rigorously? 

Reviewer #1: Yes

Reviewer #2: (No Response)

4. Have the authors made all data underlying the findings in their manuscript fully available?

Reviewer #1: Yes

Reviewer #2: Yes

5. Is the manuscript presented in an intelligible fashion and written in standard English?

Reviewer #1: Yes

Reviewer #2: Yes

6. Review Comments to the Author

Reviewer #1: I would like to congratulate the authors on their work and for the clear explanations they have provided!

Reviewer #2: (No Response)

7. PLOS authors have the option to publish the peer review history of their article (what does this mean?). If published, this will include your full peer review and any attached files.

Reviewer #1: **Yes: **Robert Ancuceanu

Reviewer #2: **Yes: **Noor Kifah Al-Tameemi

---

## [Editor Report · Acceptance letter]

11 Oct 2024

PONE-D-24-14994R1 

PLOS ONE

Dear Dr. Wu, 

I'm pleased to inform you that your manuscript has been deemed suitable for publication in PLOS ONE. Congratulations! Your manuscript is now being handed over to our production team.

Kind regards, 

on behalf of

Dr. Yong-Hong Kuo 

Academic Editor

PLOS ONE